# Investigation of the active ingredients and pharmacological mechanisms of *Porana sinensis* Hemsl. Against rheumatoid arthritis using network pharmacology and experimental validation

Jing Hu[1]☯, Lintao Zhao[1]☯, Ning Li[1]*, Yuanyuan Yang[2], Tong Qu[1], Hui Ren[1], Xiaomin Cui[1], Hongxun Tao[3], Zhiyong Chen[1]*, Yu Peng[4]*

1 Institute of Traditional Chinese Medicine, Shaanxi Academy of Traditional Chinese Medicine, Xi'an, China, 2 Department of Traditional Chinese Medicine, Xi'an Institute for Food and Drug Control, Xi'an, China, 3 School of Food and Biological Engineering, Jiangsu University, Zhenjiang, China, 4 Jiangsu Provincial Key Laboratory of Cardiovascular and Cerebrovascular Medicine, School of Pharmacy, Nanjing Medical University, Nanjing, China

☯ These authors contributed equally to this work.
* gsshxln@163.com (NL); chenzhiyong0612@sina.com (ZC); carlylepeng@gmail.com (YP)

**Data Availability Statement:** All relevant data are within the manuscript and its Supporting Information files.

## Abstract

### Background

*Porana sinensis* Hemsl. has been widely used as a substitute for Erycibes Caulis to treat rheumatoid arthritis (RA) in traditional Chinese medicine (TCM). However, little is known about the active ingredients and pharmacological mechanisms that mediate the action of *P. sinensis* against RA.

### Methods

The compounds contained in *P. sinensis* were analyzed by Q Exactive Focus mass spectrometer. The active constituents and pharmacological mechanism of *P. sinensis* against RA were clarified using a network pharmacology-based investigation. LPS-induced RAW 264.7 cells was used to verify anti-inflammatory effects of the active compounds screened by network pharmacology. Collagen-induced arthritis model was used to further investigate the mechanism of *P. sinensis* against RA.

### Results

The potential components and targets of *P. sinensis* against RA were analyzed using network pharmacology, and five compounds, twenty-five targets, and eight pathways were identified. Experimental validation suggested that *P. sinensis* extract and five compounds (esculetin, umbelliferone, *trans-N*-feruloyltyramine, caffeic acid and scopolin) could inhibit the release of inflammatory mediators (NO, TNF-α, IL-1β and IL-6) in LPS-induced RAW 264.7 cell. *P. sinensis* extract attenuated the severity, pathological changes, and release of

**Funding:** This work was financially supported by the National Natural Science Foundation of China [grant numbers 81973419, 81603264]; Key Research and Development Program of Shaanxi [grant number 2020SF-328]; Shaanxi Administration of Traditional Chinese Medicine Projects [grant number 2021-PY-003]. The funders had no role in study design, data collection and analysis, decision to publish, or preparation of the manuscript.

**Competing interests:** The authors have declared that no competing interests exist.

cytokines (IL-6 and HIF-1α) during RA progression by regulating the PI3K/AKT and HIF-1 pathways.

## Conclusion

The study provides a basis for the application of *P. sinensis* against RA. Our findings may provide suggestions for developing *P. sinensis* into a substitute for Erycibes Caulis.

## 1. Introduction

With the increasing demand for traditional Chinese medicine (TCM), some natural medicinal resources are on the verge of extinction. Developing substitutes may alleviate the pressure on these endangered Chinese medicine resources [1,2]. Erycibes Caulis is widely used in treating rheumatoid arthritis (RA) in TCM [3–5]. The official species of Erycibes Caulis recorded in China Pharmacopoeia 2020 edition are *Erycibe obtusifolia* Benth. and *Erycibe schmidtii* Craib. During the course of reduction of natural sources of Erycibes Caulis, *Porana sinensis* Hemsl. has become the primary substitute on the market [3,6]. *P. sinensis*, which belongs to the family Convolvulaceae, is mainly found in limestone mountainous regions and is widely distributed in China North-Central, China South-Central, China Southeast and Vietnam [7].

Our previous studies indicated that the chemical composition of *P. sinensis* was similar to that of Erycibes Caulis, and they all contained large amounts of coumarins and quinic acid derivatives [8]. The 40% ethanolic extract of *P. sinensis* was almost non-toxic (oral administration, 5 g/kg) and exhibited anti-inflammatory and anti-nociceptive effects [3]. Xue et al. reported that 80% methanol extract of *P. sinensis* reduced the production of NO on LPS-stimulated RAW 264.7 cells, and dicaffeoylquinic acids were the active compounds [7]. Although considerable evidence exists to support *P. sinensis* used as a substitute for Erycibes Caulis, much work is still needed to demonstrate this. For example, Erycibes Caulis is widely used in treating RA in TCM; however, there has been no study of the efficacy and mechanism of *P. sinensis* for the treatment of RA.

RA is a chronic autoimmune disease, which mainly acts on synovium, cartilage and bone, resulting in the decline of physical function and quality of life [9]. At present, nonsteroidal anti-inflammatory drugs (NSAIDs) and disease-modifying anti-rheumatic drugs (DMARDs) are commonly used in the treatment of RA. Although these drugs are typically effective, they are also not satisfactory because of their low efficacy and side effects [9]. It is of great significance to develop anti-RA TCM with multi-target effect and clear pharmacological effect.

With the rapid development of bioinformatics, network pharmacology has become a hot area of pharmacology research. Because network pharmacology delivers systematic understanding of multi-component and multi-target actions, it helps clarify the effect of TCM on various diseases [10–12]. Thus, we investigated the active constituents and pharmacological mechanism of *P. sinensis* against RA by integrating network pharmacology with experimental validation.

In this work, the chemical profile of *P. sinensis* was analyzed by Q Exactive Focus mass spectrometer (MS), and 21 compounds were identified. Subsequently, these compounds were used as the basis for network pharmacological analysis. Protein-protein interaction (PPI) network for RA was established to identify potential drug targets. KEGG pathway analysis was then performed to elucidate the signaling pathway regulated by *P. sinensis*. The results of cell experiment suggested that *P. sinensis* extract and five compounds (esculetin, umbelliferone, *trans-N-*

feruloyltyramine, caffeic acid and scopolin) could inhibit the release of inflammatory mediators (NO, TNF-α, IL-1β and IL-6) in LPS-induced RAW 264.7 cell. Animal experiments showed that *P. sinensis* treated RA by modulating PI3K-Akt and HIF-1 pathways, regulating cytokine release (IL-6 and HIF-1α). The experimental results are consistent with those of network pharmacology. This study systematically explains the effective substances and mechanisms of *P. sinensis* against RA. Our findings provide feasible suggestions for developing *P. sinensis* into a substitute for Erycibes Caulis. We also expect that the research on *P. sinensis* will accelerate the rational development and utilization of plants of the genus *Porana*.

## 2. Materials and methods

### 2.1. Materials and instrumentation

Caulis of *P. sinensis* was purchased from Kunming, China, in December 2018 (lot number: 20181205). The material was identified by Dr. Zhiyong Chen and deposited at the Shaanxi Academy of Traditional Chinese Medicine. Scopolin (lot number: 16040805), scopoletin (161208), chlorogenic acid (1701904), cryptochlorogenic acid (17061401), neochlorogenic acid (17062003), 3,5-dicaffeoylquinic acid (19061201), 3,4-dicaffeoylquinic acid (17121201), 4,5-dicaffeoylquinic acid (18070401), umbelliferone (18010202), esculetin (18092803) and caffeic acid (17122804) were bought from Qiming Bioengineering Institute (Shanghai, China), *trans-N*-feruloyltyramine (W01D9Z76497) was bought from Yuanye Bioengineering Institute (Shanghai, China), and the purities of the reference compounds were all above 98%. Bovine type II collagen (20021, Chondrex, USA), Incomplete Freund's adjuvant (F5506-10ML, Sigma −Aldrich, USA). IL-1β, IL-6 and TNF-α ELISA kits were obtained from Cloud-clone Corporation (Wuhan, China). HIF-1α ELISA kits was supplied by Jiancheng Bioengineering Institute (Nanjing, China). The RAW 264.7 murine macrophage were obtained from Youersheng Bioengineering Institute (Wuhan, China). The following antibodies were used: Actin antibody (Mouse, Servicebio, Wuhan, China), PI3K antibody (Mouse, Bioss, Beijing, China), AKT antibody (Rabbit, Servicebio, Wuhan, China), p-AKT antibody (Rabbit, Affinity, USA), and HIF-1α antibody (Rabbit, Abcam, UK).

Q Exactive Focus MS (Thermo Finnigan, San Jose, USA) was applied to identify the compounds. Samples were separated on Accucure aQ $C_{18}$ columns (2.1 mm × 150 mm, 2.6 μm) purchased from Thermo Fisher Scientific, USA. Network Pharmacology database and analysis platform: TTD (http://db.idrblab.net/ttd/); UniProt (http://www.uniprot.org/); OMIM (https://www.omim.org/); DrugBank (https://www.drugbank.ca/); GeneCards (https://www.genecards.org/); SwissTargetPrediction (http://www.swisstargetprediction.ch/); STRING (https://string-db.org/); DAVID (https://david.ncifcrf.gov/tools.jsp); R language (Version 4.0.2, https://www.r-project.org/).

### 2.2. Identification of compounds by LC-MS

**2.2.1. Standard solutions and sample preparation.** All reference compounds (1 μg/mL) were prepared in 60% methanol. Approximately 0.5 g *P. sinensis* powder (40 mesh) was extracted with 50 mL 80% methanol for 30 min by ultrasound extraction. The solution was filtered and diluted with an equal volume of 40% methanol. The sample was then filtered through 0.22-μm pore membrane.

**2.2.2. Analytical conditions.** Column temperature: 35˚C; flow rate: 0.3 mL/min; injection volume: 2 μL. A linear gradient elution of 0.1% formic acid aqueous (A) and methanol (B) was used. The elution program was optimized as follows: 5–25% B within 12 min, 25–38% B with over 12–20 min, 38–60% B with the range of 20–35 min. The MS parameters were optimized: spray voltage: ±3500 V; atomization temperature: 350˚C; capillary temperature: 320˚C; sheath

gas pressure: 45 arb; aux gas pressure, 15 arb; S-lens RF, 60 V; scan mode: full MS (resolution 70000) and MS/MS (17500).

**2.2.3. LC-MS data analysis.**    The elemental compositions were calculated according to the high-precision precursor ions. All compounds reported in *P. sinensis* and *Porana* species plants were summarized to find the most reasonable molecular formula by searching literature sources. The fragmentation patterns of these compounds were used to differentiate compounds with the same formula.

## 2.3. Network pharmacology research of *P. sinensis* against RA

**2.3.1. Targets prediction and screening.**    The compounds identified by UPLC-MS were used as the basis for network pharmacological analysis. This study predicted drug targets in databases such as TCMSP and GeneCards. Target prediction (TCMSP) was performed using the WES (Weighted Ensemble Similarity) model [13,14], which showed good performance with a consistency (82.83%), sensitivity (81.33%), and specificity (93.62%). We also predicted the constituent Target via the Swiss Target Prediction network server based on 2D and 3D similarity measures of known ligands [15]. The predicted targets were collated and imported into the UniProt database. Then, the mapping analysis of the normalized targets and the RA-related targets information obtained from databases (GeneCards, OMIM, TTD, and Drug-Bank) were conducted to screen out the potential anti-RA targets of *P. sinensis*. Finally, a component-target-disease (C-T-D) network was visualized using Cytoscape 3.6.0 plotted as an interaction network. The degree centrality (DC), betweenness centrality (BC), and closeness centrality (CC) were analyzed for each node in the C-T-D network using Cytoscape 3.6.0 software. The nodes with a DC, BC, and CC larger than the median were identified as the potential targets.

**2.3.2. PPI network.**    The PPI network was generated by importing potential drug targets into string database. The PPI network for *P. sinensis* was then constructed using Cytoscape 3.6.0. Simultaneously, the CytoHubba plug [16] in the software was utilized to screen the hub genes, and the "Degree" algorithm was selected.

**2.3.3. KEGG analysis and network construction.**    KEGG analysis for *P. sinensis* against RA was conducted using DAVID 6.8 database. The top 20 pathways with $P < 0.05$ were selected, and the R language was used for plotting. The component-target-pathway (C-T-P) network was generated using Cytoscape 3.6.0. A network analyzer was utilized for computing the topological parameters of the network. Generally, the highest-ranked target plays an essential role in anti-RA.

## 2.4. Experimental validation

**2.4.1. Preparation of *P. sinensis* extract.**    The powder of 2 kg dry samples was extracted with 20 L 40% ethanol for 2 h by reflux extraction two times. The filtered solution was concentrated using a rotary evaporator at 50°C. The yield of *P. sinensis* (Pse) was 13.0%. Eight active compounds in Pse were determined by HPLC according to our previous work [17], and the contents are 13.4268 mg/g (neochlorogenic acid), 12.6935 mg/g (scopolin), 48.5457 mg/g (chlorogenic acid), 8.2953 mg/g (cryptochlorogenic acid), 20.9330 mg/g (scopoletin), 28.6063 mg/g (3,4-dicaffeoylquinic acid), 13.5660 mg/g (3,5-dicaffeoylquinic acid) and 18.3498 mg/g (4,5-dicaffeoylquinic acid), respectively.

**2.4.2. Effects on LPS-induced RAW 264.7 cell.** *2.4.2.1. Cell culture and viability assay.* RAW 264.7 cell line was cultured in DMEM medium supplemented with 10% fetal bovine serum, and were maintained at 37°C in a water-saturated 5% $CO_2$ incubator. Cells in the mid-log phase were used for further experiments. Lipopolysaccharide (LPS, 1 μg/mL) was applied

onto RAW 264.7 cell to trigger the inflammatory responses for 24 h. Methotrexate (MTX) was used as anti-inflammatory positive control.

Cell viability was measured by MTT assay (Sigma). The RAW 264.7 cells were cultured in 96-well plate at a density of $10^4$ cells/well. Different concentrations of Pse and compounds were added 2 h before LPS treatment. After 24 h, 20 μL MTT (5 mg/mL) was added in each wells, and the cells continued to be incubated for 4 h. Then the formazan crystals were dissolved in DMSO and measured at 490 nm. Relative cell viability was calculated by comparing with that of the control group.

*2.4.2.2. Determination of pro-inflammatory cytokines.* The RAW 264.7 cells was prepared with the same procedure described above, then the cells were treated with samples (MTX, Pse, umbelliferone, caffeic acid: 120 μg/mL; scopolin, esculetin, *trans-N*-feruloyltyramine: 5 μg/mL). The supernatant was taken 24 h after administration and detected according to the instructions of NO, TNF-α, IL-1β and IL-6 kits.

**2.4.3. Effects on collagen-induced arthritis model.**   *2.4.3.1. Animals.* The study protocol of animal experiments was reviewed and approved by the Experimental Animal Ethical Committee at the Shaanxi Academy of Traditional Chinese Medicine (license numbers AF/SL-01/01.2 and AF/SC-05/01.2). All rats received human care followed the National Institutes of Health Guide for the Care and Use of Laboratory Animals (NIH Publications No. 8023). Male SD rats (SPF grade, 180–220 g) were brought from the Experimental Animal Facilities of Xi'an Jiaotong University (SYXK2020-005). The animals were kept with a 12/12 h light/dark cycle at a temperature of 22 ± 1˚C and 65 ± 5% humidity.

*2.4.3.2. Induction of collagen-induced arthritis (CIA).* The 48 rats were randomly divided into six groups: the normal, model, methotrexate (1 mg/kg), high-dose (Pse, 0.6 g/kg), middle-dose (Pse, 0.3 g/kg) and low-dose (Pse, 0.15 g/kg) groups. The selection of administered doses is based on our previous studies. At these doses, the extract shows good anti-inflammatory and analgesic effects [3]. The CIA model is one of the standard RA models, which shares several pathological features with RA, such as synovial inflammatory cell infiltration, synovial hyperplasia and bone erosion [9]. The CIA model was established as previously described [18]. Type II collagen was prepared with 0.1 mol/L acetic acid at 2 mg/mL. Then, the solution was emulsified with an equal volume of incomplete Freund's adjuvant. For immunization, 0.2 mL of collagen was injected at the base of the tail of each rat. The rats were given a booster injection (0.2 mL of collagen) on day 7. On day 13, Pse (0.6, 0.3, 0.15 g/kg) was intragastric administrated once daily for 17 days, while methotrexate was given every 3 days.

During the experiment, body weights were measured every 7 days. The left hind paw's ankle circumference and the arthritis index (AI) were recorded every 4 days from day 13. The AI criteria [19]: 0, no swelling; 1, swelling or redness of digit; 2, slight swelling of the ankle; 3, gross swelling of the paw; and 4, severe arthritis of the entire paw. Hind paws were used to calculate AI. Relative organ weights were assayed as well.

*2.4.3.3. Biochemical assays.* On day 29, all rats were anesthetized with pentobarbital sodium, and the abdominal aorta was punctured for blood collection. The blood was centrifuged to separate the serum for ELISA testing of HIF-1α and IL-6 using assay kits.

*2.4.3.4. Histopathological examination.* The right hind ankle joints were harvested after serum collection, fixed in 10% formalin, and decalcified in ethylenediaminetetraacetic acid. The joints were then embedded in paraffin, stained with hematoxylin and eosin (H&E), and observed under light microscope.

*2.4.3.5. Quantitative PCR analysis.* The mRNA was extracted using the TRIzol reagent from the synovium of the joint. For complementary DNA (cDNA) synthesis, reverse transcriptase and 2 μg RNA were used. PCR amplification was conducted using gene-specific PCR primers provided by Wuhan Servicebio Technology (Wuhan, China). Primers used in this study are

listed in **S1 Table**. The amplification was performed in a 15 μL reaction volume containing 2 μL cDNA, 1.5 μL 2.5 μmol/L primers, 7.5 μL 2 × qPCR mix, and 4 μL ddH$_2$O. Each reaction was carried out for 40 cycles of denaturation at 95˚C for 15 s and annealing at 60˚C for 60 s. GAPDH was used as a control for normalization.

*2.4.3.6. Western blot analysis.* The experiment was performed following previous studies' protocol with slight modification [9]. The protein extraction of the synovium of joints was performed with RIPA lysis buffer. The lysate was then centrifuged at 4˚C, 12000 rpm for 10 min. Subsequently, the supernatant was collected and boiled for 15 min, subjected to SDS-PAGE, and transferred to PVDF membrane. After incubating with respective primary antibodies, the membrane was extensive washed with TBST, and treated with horseradish peroxidase-conjugated secondary antibodies. In the experiment, β-actin (GB12001, Wuhan Servicebio Technology) was regarded as the internal control. Protein visualization was conducted on an imaging system (Clinx, Shanghai, China).

## 2.5. Statistical analysis

All data were expressed as mean ± SD for each group. Comparisons between multiple groups were made with one-way ANOVA, and a value of $p < 0.05$ was considered statistically significant.

## 3. Results

### 3.1. LC-MS analysis of *P. sinensis*

The total ion chromatograms are shown in **Fig 1**. A total of 21 compounds and isomers from *P. sinensis* were identified, including seven coumarins (5, 8, 10, 11, 12, 13, 15), seven chlorogenic acids (2, 4, 6, 9, 16, 17, 20), one tropane alkaloid (1), two amides (19, 21), one flavonoid (18), one lignan (14) and two other compounds (3, 7). Twelve compounds were unambiguously identified as neochlorogenic acid (2), chlorogenic acid (4), scopolin (5), cryptochlorogenic acid (6), caffeic acid (7), umbelliferone (8), scopoletin (10), esculetin (15), 3,4-dicaffeoylquinic acid (16), 3,5-dicaffeoylquinic acid (17), 4,5-dicaffeoylquinic acid (20),

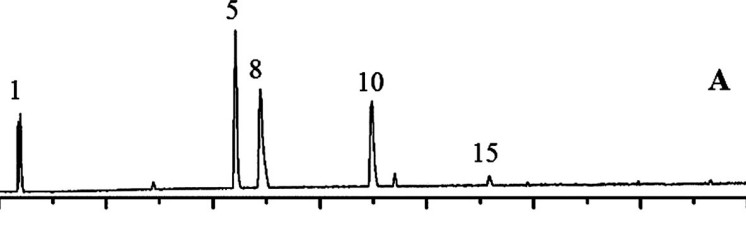

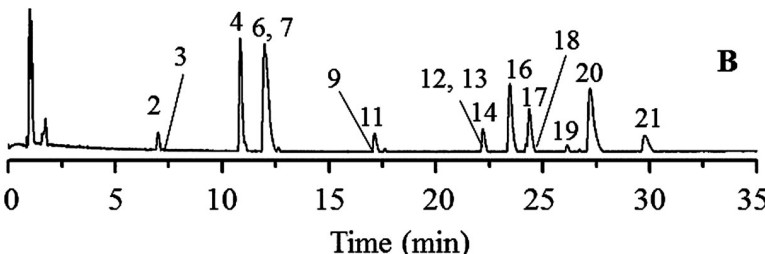

**Fig 1.** Total ion chromatograms of *P. sinensis* in positive ion (A) and negative ion (B) modes.

**Table 1. Identification of chemical constituents from *P. sinensis* by UPLC-MS.**

| No. | $t_R$ (min) | Observed *m/z* | Calculated *m/z* | Mode | Error (ppm) | Molecular formula | Fragment ions (*m/z*) | Identification | References |
|---|---|---|---|---|---|---|---|---|---|
| 1 | 0.97 | 144.1015 | 144.1019 | [M+H]$^+$ | -2.77 | $C_7H_{13}NO_2$ | 144.1015, 126.0910, 108.0806, 98.0962, 84.0806, 68.0696 | baogongteng C or erycibelline | [17] |
| 2* | 7.00 | 353.0862 | 353.0868 | [M-H]$^-$ | -1.70 | $C_{16}H_{18}O_9$ | 353.0859, 191.0546, 179.0334, 173.0443, 161.0227, 135.0435 | neochlorogenic acid | [7,8] |
| 3 | 7.28 | 153.0177 | 153.0182 | [M-H]$^-$ | -3.27 | $C_7H_6O_4$ | 153.0178, 141.9110, 123.0074, 109.0278, 61.9858 | 3,4-dihydroxybenzoic acid | [7,8] |
| 4* | 10.86 | 353.0862 | 353.0868 | [M-H]$^-$ | -1.70 | $C_{16}H_{18}O_9$ | 191.0560, 179.0347, 161.0242 | chlorogenic acid | [7,8] |
| 5* | 11.04 | 355.1014 | 355.1024 | [M+H]$^+$ | -2.82 | $C_{16}H_{18}O_9$ | 193.0493, 178.0259, 165.0543, 133.0282 | scopolin | [7,8] |
| 6* | 11.98 | 353.0862 | 353.0868 | [M-H]$^-$ | -1.70 | $C_{16}H_{18}O_9$ | 191.0545, 179.0347, 173.0452, 135.0434 | cryptochlorogenic acid | [7,8] |
| 7* | 12.07 | 179.0334 | 179.0339 | [M-H]$^-$ | -2.79 | $C_9H_8O_4$ | 179.0334, 135.0435 | caffeic acid | [7] |
| 8* | 12.21 | 163.0383 | 163.0388 | [M+H]$^+$ | -3.07 | $C_9H_6O_3$ | 163.0385, 145.0281, 135.0437, 117.0332, 107.0490, 89.0384 | umbelliferone | [20] |
| 9 | 17.13 | 367.1017 | 367.1024 | [M-H]$^-$ | -1.91 | $C_{17}H_{20}O_9$ | 191.0545, 173.0438, 135.0356, 111.0435, 93.0329, 87.0071 | methyl chlorogenate | [21] |
| 10* | 17.44 | 193.0492 | 193.0495 | [M+H]$^+$ | -1.55 | $C_{10}H_8O_4$ | 193.0491, 178.0255, 165.0539, 149.0591, 137.0595, 133.0281 | scopoletin | [3] |
| 11 | 19.85 | 695.1804 | 695.1818 | [M-H]$^-$ | -2.01 | $C_{31}H_{36}O_{18}$ | 359.0966, 335.0766, 197.0441, 173.0437, 153.0538, 135.0438 | eryciboside F | [21] |
| 12 | 21.60 | 665.1702 | 665.1712 | [M-H]$^-$ | -1.50 | $C_{30}H_{34}O_{17}$ | 198.0475, 191.0334, 176.0099, 153.0540, 121.0278 | eryciboside A or B or C | [21] |
| 13 | 21.75 | 635.1597 | 635.1607 | [M-H]$^-$ | -1.57 | $C_{29}H_{32}O_{16}$ | 191.0334, 167.0334, 121.0433 | eryciboside D or E | [21] |
| 14 | 22.19 | 595.2012 | 595.2021 | [M-H]$^-$ | -1.51 | $C_{28}H_{36}O_{14}$ | 447.4385, 433.1486, 418.1223, 403.1426, 373.1277, 358.1036, 239.7386, 181.0488, 139.0023 | aketrilignoside B | [21] |
| 15* | 22.92 | 179.0333 | 179.0338 | [M+H]$^+$ | -2.79 | $C_9H_6O_4$ | 179.0333, 151.0386, 133.0280, 123.0436 | esculetin | [22] |
| 16* | 23.46 | 515.1171 | 515.1184 | [M-H]$^-$ | -2.52 | $C_{25}H_{24}O_{12}$ | 353.0859, 335.0760, 191.0546, 179.0334, 173.0439, 161.0228, 135.0435, 93.0329 | 3,4-dicaffeoylquinic acid | [7,8] |
| 17* | 24.38 | 515.1171 | 515.1184 | [M-H]$^-$ | -2.52 | $C_{25}H_{24}O_{12}$ | 353.0862, 191.0545, 179.0333, 173.0437, 135.0434 | 3,5-dicaffeoylquinic acid | [7,8] |
| 18 | 24.67 | 287.0548 | 287.0550 | [M-H]$^-$ | -0.697 | $C_{15}H_{12}O_6$ | 243.0648, 199.0746, 177.0539, 163.0380, 137.0223, 119.0485, 93.0330, 61.9869 | eriodictyol | [21] |
| 19 | 25.87 | 282.1123 | 282.1125 | [M-H]$^-$ | -0.709 | $C_{17}H_{17}O_3N$ | 282.1122, 197.9011, 162.0543, 1485.0278, 136.0750, 119.0485 | *trans-N*-(p-coumaroyl) tyramine | [20] |
| 20* | 27.23 | 515.1171 | 515.1184 | [M-H]$^-$ | -2.52 | $C_{25}H_{24}O_{12}$ | 353.0858, 191.0545, 179.0333, 173.0438, 135.0434, 93.0327 | 4,5-dicaffeoylquinic acid | [7,8] |
| 21* | 29.85 | 312.1246 | 312.1238 | [M-H]$^-$ | 2.56 | $C_{18}H_{19}O_4N$ | 312.1238, 297.1001, 190.0508, 178.0506, 148.0528, 135.0451 | *trans-N*-feruloyltyramine | [20] |

* Compared with reference compounds.

and *trans-N*-feruloyltyramine (21) by comparisons with reference compounds. Data for all compounds are listed in **Table 1**.

## 3.2. Network pharmacology analysis of *P. sinensis*

### 3.2.1. C-T-D network construction and core target screening.
Through Swiss Target Prediction and DrugBank reverse Prediction, 466 potential targets for 21 chemical components were obtained. The 466 potential targets were mapped to RA targets, and 293 common potential targets against RA were obtained. Using these targets as network nodes, Cytoscape

**Table 2. The topological parameter analysis of 10 compounds in *P. sinensis* against rheumatoid arthritis.**

| No. | compounds | molecular formula | Betweenness centrality (BC) | Closeness centrality (CC) | Degree centrality (DC) |
|---|---|---|---|---|---|
| 1 | Scopolin | $C_{16}H_{18}O_9$ | 0.00554748 | 0.3665 | 45 |
| 2 | Scopoletin | $C_{10}H_8O_4$ | 0.01196908 | 0.3842 | 86 |
| 3 | Neochlorogenicacid | $C_{16}H_{18}O_9$ | 0.00316416 | 0.3642 | 40 |
| 4 | 4,5-Dicaffeoylquinicacid | $C_{25}H_{24}O_{12}$ | 0.00264869 | 0.3639 | 44 |
| 5 | 3,5-Dicaffeoylquinicacid | $C_{25}H_{24}O_{12}$ | 0.00366490 | 0.3639 | 42 |
| 6 | 3,4-Dicaffeoylquinicacid | $C_{25}H_{24}O_{12}$ | 0.00264869 | 0.3639 | 44 |
| 7 | Esculetin | $C_9H_6O_3$ | 0.02573882 | 0.4049 | 116 |
| 8 | Caffeic acid | $C_9H_8O_4$ | 0.01098960 | 0.3787 | 64 |
| 9 | Umbelliferone | $C_9H_6O_3$ | 0.01219321 | 0.3861 | 81 |
| 10 | *trans-N*-feruloyltyramine | $C_{18}H_{19}O_4N$ | 0.02687441 | 0.4122 | 102 |

3.6.0 software was used to analyze network topology parameters. The results showed that the median DC was 41, the average BC was 0.0026, and the average CC was 0.3639 for the compounds; the median DC was 2, the average BC was 0.0013, and the average CC was 0.4897 for the 293 common targets. There were ten compounds and 112 common targets with DC, BC, and CC values higher than the median, shown in **S1 Fig**, **Tables 2** and **3**.

**3.2.2. Construction of the PPI network.** Searching for proteins playing important roles in the PPI network, we uploaded the 112 anti-RA targets to the STRING database to observe these protein interaction relationships. Then, we imported the obtained TSV file into Cytoscape 3.6.0 to build the PPI network. Using the Cytohubba plugin, the greater the node's degree value, the redder its color, the more important it is in the network (**S2 Fig**). The hub genes include CAPDH (Degree = 92), AKT1 (Degree = 91), GASP3 (Degree = 81), EGFR (Degree = 78), SRC (Degree = 76), HSP90AA1 (Degree = 75), MAPK1 (Degree = 72), TNF (Degree = 67), ESR1 (Degree = 67), STAT3 (Degree = 66).

**Table 3. The topological parameter analysis of 112 targets with rheumatoid arthritis for *P. sinensis*.**

| NO. | UniProt | Gene names | Betweenness centrality (BC) | Closeness centrality (CC) | Degree centrality (DC) |
|---|---|---|---|---|---|
| 1 | Q14790 | CASP8 | 0.00441081 | 0.49481865 | 6 |
| 2 | P08246 | ELANE | 0.00673174 | 0.49739583 | 8 |
| 3 | P08183 | ABCB1 | 0.00741662 | 0.49739583 | 8 |
| 4 | P42574 | CASP3 | 0.00719441 | 0.49739583 | 8 |
| 5 | P17252 | PRKCA | 0.00719441 | 0.49739583 | 8 |
| 6 | Q05655 | PRKCD | 0.00541423 | 0.4961039 | 7 |
| 7 | P08253 | MMP2 | 0.01899941 | 0.5 | 10 |
| 8 | P05067 | APP | 0.00673172 | 0.49739583 | 8 |
| 9 | P15121 | AKR1B1 | 0.02883723 | 0.50395778 | 13 |
| 10 | P21964 | COMT | 0.00114453 | 0.49226804 | 4 |
| 11 | P56817 | BACE1 | 0.01344752 | 0.49869452 | 9 |
| 12 | P04626 | ERBB2 | 0.00064866 | 0.49100257 | 3 |
| 13 | P09874 | PARP1 | 0.00114453 | 0.49226804 | 4 |
| 14 | P03372 | ESR1 | 0.00200793 | 0.49354005 | 5 |
| 15 | P27338 | MAOB | 0.00224914 | 0.49354005 | 5 |
| 16 | P05177 | CYP1A2 | 0.00144712 | 0.49226804 | 4 |
| 17 | P22303 | ACHE | 0.00078702 | 0.49100257 | 3 |

*(Continued)*

**Table 3.** (Continued)

| NO. | UniProt | Gene names | Betweenness centrality (BC) | Closeness centrality (CC) | Degree centrality (DC) |
|---|---|---|---|---|---|
| 18 | P35968 | KDR | 0.00570423 | 0.4961039 | 7 |
| 19 | P12931 | SRC | 0.00247146 | 0.49354005 | 5 |
| 20 | P00390 | GSR | 0.00114453 | 0.49226804 | 4 |
| 21 | P49841 | GSK3B | 0.00114453 | 0.49226804 | 4 |
| 22 | P09769 | FGR | 0.00114453 | 0.49226804 | 4 |
| 23 | P0DMV8 | HSPA1A | 0.00114453 | 0.49226804 | 4 |
| 24 | P53350 | PLK1 | 0.00114453 | 0.49226804 | 4 |
| 25 | Q05397 | PTK2 | 0.00064866 | 0.49100257 | 3 |
| 26 | P09917 | ALOX5 | 0.00154686 | 0.49226804 | 4 |
| 27 | Q92731 | ESR2 | 0.00200793 | 0.49354005 | 5 |
| 28 | P42262 | GRIA2 | 0.00032983 | 0.48974359 | 2 |
| 29 | P07550 | ADRB2 | 0.00091322 | 0.49100257 | 3 |
| 30 | P00533 | EGFR | 0.00455135 | 0.4961039 | 7 |
| 31 | P28482 | MAPK1 | 0.00130213 | 0.49100257 | 3 |
| 32 | Q07820 | MCL1 | 0.00067511 | 0.48974359 | 2 |
| 33 | P04406 | GAPDH | 0.00067511 | 0.48974359 | 2 |
| 34 | P01375 | TNF | 0.00116211 | 0.49100257 | 3 |
| 35 | P11142 | HSPA8 | 0.00067511 | 0.48974359 | 2 |
| 36 | P20248 | CCNA2 | 0.00234162 | 0.49354005 | 5 |
| 37 | P78396 | CCNA1 | 0.00234162 | 0.49354005 | 5 |
| 38 | P24941 | CDK2 | 0.00334262 | 0.49481865 | 6 |
| 39 | P14635 | CCNB1 | 0.00116211 | 0.49100257 | 3 |
| 40 | P06493 | CDK1 | 0.00116211 | 0.49100257 | 3 |
| 41 | P11802 | CDK4 | 0.00334262 | 0.49481865 | 6 |
| 42 | P24385 | CCND1 | 0.00234162 | 0.49354005 | 5 |
| 43 | P11021 | HSPA5 | 0.00067511 | 0.48974359 | 2 |
| 44 | P60568 | IL2 | 0.00067511 | 0.48974359 | 2 |
| 45 | P19367 | HK1 | 0.00067511 | 0.48974359 | 2 |
| 46 | P52789 | HK2 | 0.00067511 | 0.48974359 | 2 |
| 47 | P30542 | ADORA1 | 0.01223823 | 0.49481865 | 6 |
| 48 | P0DMS8 | ADORA3 | 0.0151816 | 0.49739583 | 8 |
| 49 | P10275 | AR | 0.00067511 | 0.48974359 | 2 |
| 50 | P35869 | AHR | 0.00111812 | 0.49100257 | 3 |
| 51 | P35354 | PTGS2 | 0.00546993 | 0.4961039 | 7 |
| 52 | P18031 | PTPN1 | 0.00140827 | 0.4961039 | 7 |
| 53 | Q02750 | MAP2K1 | 0.00184084 | 0.49100257 | 3 |
| 54 | P29323 | EPHB2 | 0.00028174 | 0.48974359 | 2 |
| 55 | P04818 | TYMS | 0.00028174 | 0.48974359 | 2 |
| 56 | O14672 | ADAM10 | 0.00028174 | 0.48974359 | 2 |
| 57 | P55072 | VCP | 0.00028174 | 0.48974359 | 2 |
| 58 | P08238 | HSP90AB1 | 0.00028174 | 0.48974359 | 2 |
| 59 | P14625 | HSP90B1 | 0.00028174 | 0.48974359 | 2 |
| 60 | P50281 | MMP14 | 0.00028174 | 0.48974359 | 2 |
| 61 | P15144 | ANPEP | 0.00028174 | 0.48974359 | 2 |
| 62 | AGTR1 | AGTR1 | 0.00028174 | 0.48974359 | 2 |
| 63 | P11387 | TOP1 | 0.00023523 | 0.49100257 | 3 |
| 64 | Q13547 | HDAC1 | 0.00028174 | 0.48974359 | 2 |

(*Continued*)

**Table 3.** (Continued)

| NO. | UniProt | Gene names | Betweenness centrality (BC) | Closeness centrality (CC) | Degree centrality (DC) |
|---|---|---|---|---|---|
| 65 | P35462 | DRD3 | 0.00028174 | 0.48974359 | 2 |
| 66 | P14416 | DRD2 | 0.00028174 | 0.48974359 | 2 |
| 67 | O60760 | HPGDS | 0.00028174 | 0.48974359 | 2 |
| 68 | P23443 | RPS6KB1 | 0.00028174 | 0.48974359 | 2 |
| 69 | Q05193 | DNM1 | 0.00028174 | 0.48974359 | 2 |
| 70 | P42345 | MTOR | 0.00028174 | 0.48974359 | 2 |
| 71 | O14757 | CHEK1 | 0.00028174 | 0.48974359 | 2 |
| 72 | Q92769 | HDAC2 | 0.00034993 | 0.48974359 | 2 |
| 73 | P19838 | NFKB1 | 0.00034993 | 0.48974359 | 2 |
| 74 | P40763 | STAT3 | 0.00045826 | 0.48974359 | 2 |
| 75 | Q16236 | NFE2L2 | 0.00045826 | 0.48974359 | 2 |
| 76 | O15054 | KDM6B | 0.00045826 | 0.48974359 | 2 |
| 77 | P42336 | PIK3CA | 0.00045826 | 0.48974359 | 2 |
| 78 | P08684 | CYP3A4 | 0.00045826 | 0.48974359 | 2 |
| 79 | P42338 | PIK3CB | 0.00045826 | 0.48974359 | 2 |
| 80 | P06239 | LCK | 0.00045826 | 0.48974359 | 2 |
| 81 | O00206 | TLR4 | 0.00045826 | 0.48974359 | 2 |
| 82 | P14780 | MMP9 | 0.00329356 | 0.49226804 | 4 |
| 83 | P37840 | SNCA | 0.00066095 | 0.49100257 | 3 |
| 84 | Q14289 | PTK2B | 0.00066095 | 0.49100257 | 3 |
| 85 | P09619 | PDGFRB | 0.00066095 | 0.49100257 | 3 |
| 86 | P08069 | IGF1R | 0.00023292 | 0.48974359 | 2 |
| 87 | P07948 | LYN | 0.00066095 | 0.49100257 | 3 |
| 88 | P31749 | AKT1 | 0.00066095 | 0.49100257 | 3 |
| 89 | P11388 | TOP2A | 0.00023292 | 0.48974359 | 2 |
| 90 | P05771 | PRKCB | 0.00023292 | 0.48974359 | 2 |
| 91 | P17612 | PRKACA | 0.00023292 | 0.48974359 | 2 |
| 92 | O75469 | NR1I2 | 0.00023292 | 0.48974359 | 2 |
| 93 | P15559 | NQO1 | 0.00023292 | 0.48974359 | 2 |
| 94 | P23458 | JAK1 | 0.00023292 | 0.48974359 | 2 |
| 95 | P07900 | HSP90AA1 | 0.00128134 | 0.49226804 | 4 |
| 96 | Q99527 | GPER1 | 0.00023292 | 0.48974359 | 2 |
| 97 | P26358 | DNMT1 | 0.00023292 | 0.48974359 | 2 |
| 98 | Q16678 | CYP1B1 | 0.00023292 | 0.48974359 | 2 |
| 99 | Q00534 | CDK6 | 0.00023292 | 0.48974359 | 2 |
| 101 | P60709 | ACTB | 0.00023292 | 0.48974359 | 2 |
| 102 | P08254 | MMP3 | 0.00068484 | 0.49100257 | 3 |
| 103 | P06400 | RB1 | 0.00023292 | 0.48974359 | 2 |
| 104 | P38936 | CDKN1A | 0.00023292 | 0.48974359 | 2 |
| 105 | P10415 | BCL2 | 0.00023292 | 0.48974359 | 2 |
| 106 | P08473 | MME | 0.00074482 | 0.48974359 | 2 |
| 107 | P29317 | EPHA2 | 0.00127883 | 0.49100257 | 3 |
| 108 | P00519 | ABL1 | 0.00127883 | 0.49100257 | 3 |
| 109 | P05556 | ITGB1 | 0.00074482 | 0.48974359 | 2 |
| 110 | P16109 | SELP | 0.00141277 | 0.49100257 | 3 |
| 111 | P16581 | SELE | 0.00141277 | 0.49100257 | 3 |
| 112 | P02766 | TTR | 0.00141277 | 0.49100257 | 3 |

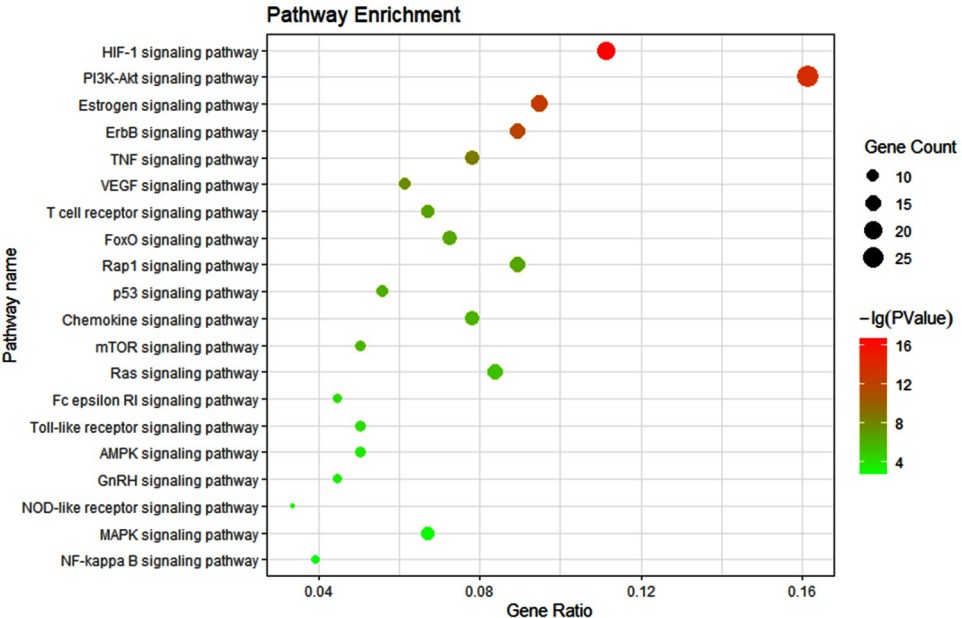

**Fig 2. KEGG pathway enrichment analysis of 112 target genes.**

**3.2.3. KEGG analysis and C-T-P construction.** The 112 targets were mapped to 122 pathways using DAVID. Unrelated pathways, such as "Pathways in cancer", "Hepatitis B" and "Bladder cancer" were excluded. The top 20 KEGG pathways were obtained based on *P*-value (**S3 Fig**, **S2 Table**), shown in **Fig 2**. A total of 64 targets were directly connected to the top 20 pathways. Most of these pathways are involved in inflammation, such as PI3K-Akt, HIF-1, ErbB, Rap1, TNF, VEGF, Ras signaling pathways.

The C-T-P network for the *P. sinensis*-mediated treatment of RA is shown in **Fig 3**. The network comprises 94 nodes (10 compounds, 64 targets, and 20 pathways) and 400 edges. The green triangle represents the compounds, the yellow diamond denotes the potential targets, and the purple V graphics represents the pathways. The topological parameters of C-T-P were

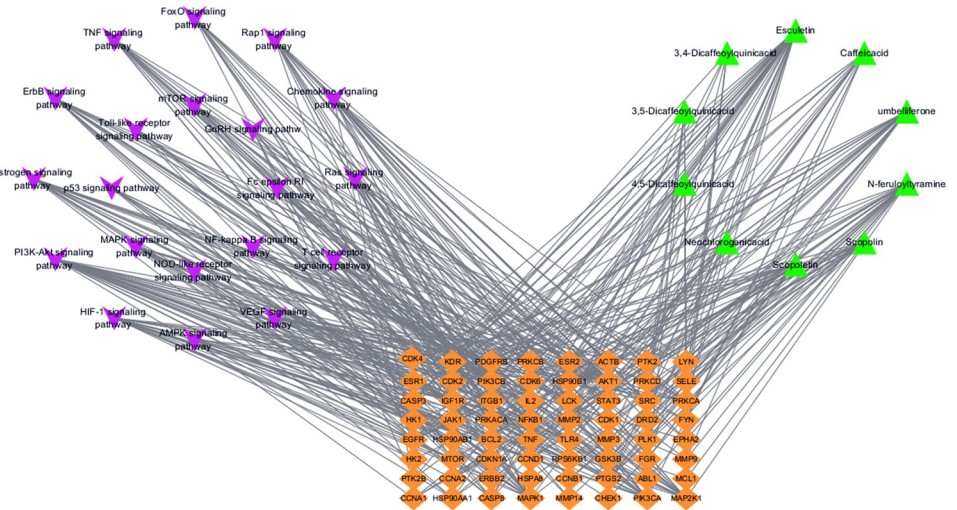

**Fig 3. The C-T-P network for *P. sinensis* in treatment of rheumatoid arthritis.**

calculated using the Network Analyzer (**Table 4**). The results show that the median DC was 16, the median BC was 0.0351, and the median CC was 0.4115 for the compounds. There were 5 compounds with DC, BC and CC values higher than the median: esculetin (DC = 30, BC = 0.1328, CC = 0.4898), umbelliferone (DC = 20, BC = 0.0481, CC = 0.4348), *trans-N*-feruloyltyramine (DC = 21, BC = 0.0917, CC = 0.4471), caffeic acid (DC = 16, BC = 0.0351, CC = 0.4115) and scopolin (DC = 16, BC = 0.0547, CC = 0.4152). The median values of DC, BC, and CC were 7, 0.0093 and 0.4155 for targets nodes, respectively. There were 25 targets that exhibited high topological values. For the pathway nodes, the median values of DC, BC

**Table 4. The topological parameter analysis of C-T-P for *P. sinensis* in treatment of rheumatoid arthritis.**

| Number | Node | Betweenness centrality (BC) | Closeness centrality (CC) | Degree centrality (DC) |
|---|---|---|---|---|
| 1 | *trans-N*-feruloyltyramine | 0.0917 | 0.4471 | 21 |
| 2 | umbelliferone | 0.0481 | 0.4348 | 20 |
| 3 | Caffeic acid | 0.0351 | 0.4115 | 13 |
| 4 | Esculetin | 0.1328 | 0.4898 | 30 |
| 5 | Scopolin | 0.0547 | 0.4152 | 16 |
| 6 | Ras signaling pathway | 0.0225 | 0.4189 | 15 |
| 7 | Chemokine signaling pathway | 0.0244 | 0.4043 | 14 |
| 8 | Rap1 signaling pathway | 0.0380 | 0.4189 | 16 |
| 9 | TNF signaling pathway | 0.0460 | 0.4189 | 14 |
| 10 | ErbB signaling pathway | 0.0275 | 0.4227 | 16 |
| 11 | Estrogen signaling pathway | 0.0531 | 0.4227 | 17 |
| 12 | PI3K-Akt signaling pathway | 0.1271 | 0.4745 | 29 |
| 13 | HIF-1 signaling pathway | 0.0593 | 0.4387 | 19 |
| 14 | MAP2K1 | 0.0379 | 0.4947 | 16 |
| 15 | RPS6KB1 | 0.0057 | 0.3974 | 6 |
| 16 | MTOR | 0.0057 | 0.3974 | 6 |
| 17 | NFKB1 | 0.0211 | 0.4471 | 11 |
| 18 | PIK3CA | 0.0331 | 0.4947 | 16 |
| 19 | PIK3CB | 0.0331 | 0.4947 | 16 |
| 20 | PDGFRB | 0.0044 | 0.4009 | 6 |
| 21 | IGF1R | 0.0064 | 0.4079 | 7 |
| 22 | AKT1 | 0.0462 | 0.5054 | 18 |
| 23 | PRKCB | 0.0060 | 0.4009 | 8 |
| 24 | PRKACA | 0.0043 | 0.3875 | 6 |
| 25 | HSP90AA1 | 0.0095 | 0.4306 | 6 |
| 26 | CDKN1A | 0.0071 | 0.4079 | 6 |
| 27 | CASP3 | 0.0195 | 0.4115 | 8 |
| 28 | PRKCA | 0.0353 | 0.4346 | 12 |
| 29 | KDR | 0.0156 | 0.4152 | 9 |
| 30 | SRC | 0.0153 | 0.4189 | 10 |
| 31 | GSK3B | 0.0068 | 0.4079 | 7 |
| 32 | EGFR | 0.0445 | 0.4895 | 15 |
| 33 | MAPK1 | 0.0558 | 0.5167 | 19 |
| 34 | TNF | 0.0185 | 0.4043 | 10 |
| 35 | CDK2 | 0.0152 | 0.4471 | 8 |
| 36 | CDK4 | 0.0172 | 0.4515 | 8 |
| 37 | CCND1 | 0.0127 | 0.4266 | 8 |
| 38 | PTGS2 | 0.0229 | 0.4387 | 9 |

and CC were 12, 0.0224 and 0.4043. There were 8 pathways that exhibited high topological values.

## 3.3. Experimental validation for *P. sinensis* against RA

**3.3.1. Effects on LPS-induced RAW 264.7 cell.**   Prior to validation of the effects of Pse and five active compounds on LPS-induced RAW 264.7 cell, we first analyzed their cellular toxicities using the MTT cell viability assay. The results showed that RAW 264.7 cell viability was 83.6% after induction with LPS (1 μg/mL) alone, so we chose this concentration for subsequent experiments. Based on the anti-inflammatory activity and cytotoxicity, we selected the dose concentration of the samples (**S4 Fig**). As shown in **Fig 4**, Pse and the five compounds screened by network pharmacology all showed good anti-inflammatory activities, and the inhibition effect of Pse on NO and IL-6 was better than umbelliferone and caffeic acid at the same concentration. Inhibitory effects of *trans-N*-feruloyltyramine on IL-1β and IL-6 were not obvious.

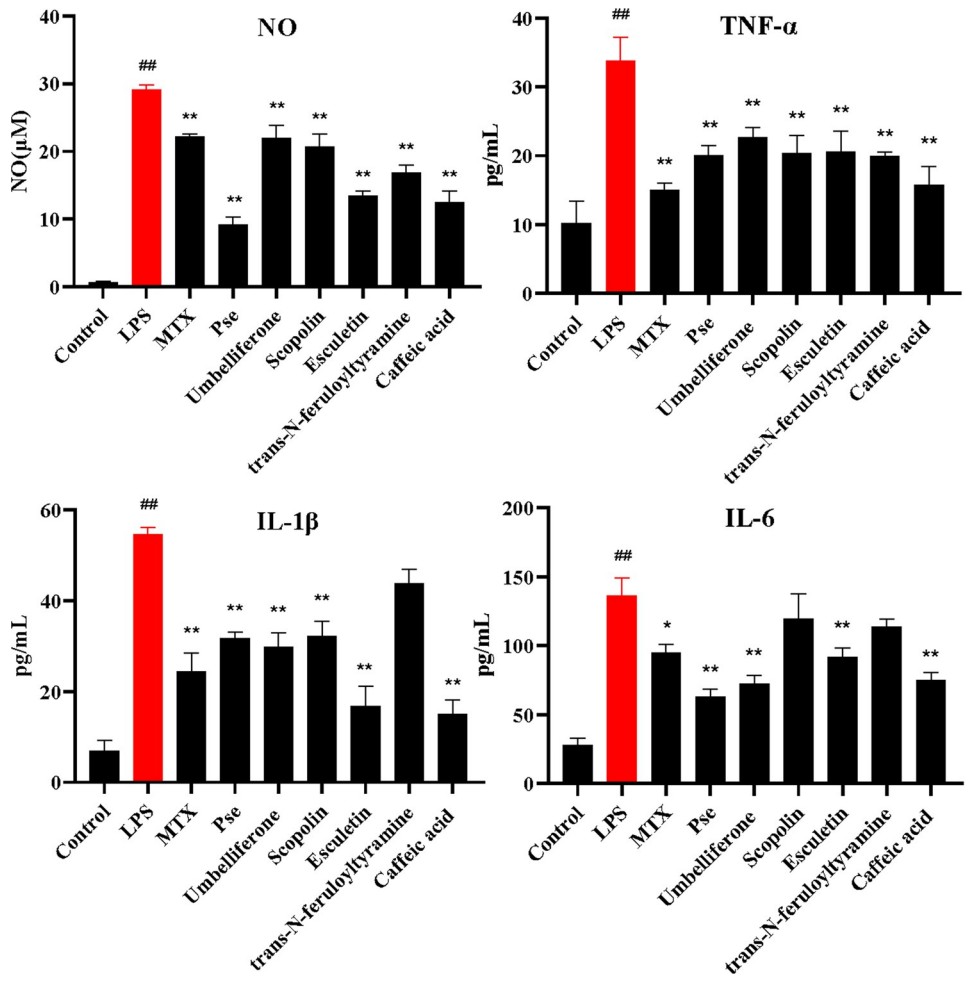

**Fig 4. Inhibitory effect of *P. sinensis* extract (Pse), methotrexate (MTX) and five compounds (esculetin, umbelliferone, *trans-N*-feruloyltyramine, caffeic acid and scopolin) on the release of inflammatory mediators (NO, TNF-α, IL-1β and IL-6) in LPS-induced RAW 264.7 cell.** $^{*}P < 0.05$ and $^{**}P < 0.01$ compared with model group; $^{\#}P < 0.05$ and $^{\#\#}P < 0.01$ compared with normal group.

**3.3.2. Effects on collagen-induced arthritis model.** *3.3.2.1. Effects of P. sinensis on ankle circumference and AI.* Red swelling in the paws of model rats was first observed on day 11 following CIA-induced modeling. As shown in **S5 Fig**, edema of the left hind paw and AI of rats were significantly different between normal and model groups on day 13, which represents the successful model. The arthritis severity in the model group reached a peak on Day 21. Pse (0.6 g/kg) can suppress the paw swelling and AI on day 29.

*3.3.2.2. Effects of P. sinensis on body weight and relative organ weight.* As shown in **S5 Fig**, body weight in normal group rats increased more than CIA rats ($p < 0.01$), while Pse treated rats (0.6 g/kg, 0.3 g/kg) did not achieve a significant difference from model group rats ($p > 0.05$). The weight gain of rats in the methotrexate treated group was less than that in the model group ($p < 0.05$). As shown in **Table 5**, the spleen's relative organ weight in methotrexate treated rats was significantly higher than rats of other groups ($p < 0.01$).

*3.3.2.3. Effect of P. sinensis on serum cytokine levels.* As shown in **S6 Fig**, collagen-induced the significant production of HIF-1α and IL-6 compared with the normal group. Pse (0.6 g/kg) inhibited the collagen-induced production of HIF-1α and IL-6.

*3.3.2.4. Effect of P. sinensis on histopathological changes.* The histopathological changes of ankle joints in each group are shown in **Fig 5**. In the model group, inflammatory infiltration, synovial hyperplasia, and cartilage degradation were observed in the ankle joint. Methotrexate and Pse significantly attenuated the synovial hyperplasia and cartilage degradation.

*3.3.2.5. Effects of P. sinensis on the mRNA expression.* As shown in **S7 Fig**, compared with the normal group, the mRNA levels of HIF-1α, PI3K, and AKT in the model group were significantly increased ($P < 0.05$). The high-dose group of Pse has an effect of decreasing the mRNA levels of HIF-1α, PI3K, and AKT ($P < 0.05$).

*3.3.2.6. Effects of P. sinensis on the protein expression levels.* As shown in **Fig 6**, the expression levels of PI3K, AKT, p-AKT, and HIF-1α proteins were significantly increased in the synovium of CIA model rats, compared to the normal group ($p < 0.01$). After the administration of Pse (0.6 g/kg), their expression levels were significantly reduced ($p < 0.05$).

## 4. Discussion

The similarity of chemical compositions is an essential indicator for judging the rationality of substitutes. In this study, a total of 21 compounds from *P. sinensis* were identified by UPLC-MS, including seven coumarins, seven chlorogenic acids, one tropane alkaloid, two amides, one flavonoid, one lignan, and two other compounds. By comparing with other studies [4,8,23,24], we discovered the chemical composition of *P. sinensis* is similar to that of *E. obtusifolia* and *E. schmidtii*. It should be noted that there are few reports on the systematic

**Table 5. Relative organ weights in rats.** Data represent the mean ± SD.

| Relative organ weight (g/100 g) | Normal | Model | Methotrexate (1 mg/kg) | Pse (0.6 g/kg) | Pse (0.3 g/kg) | Pse (0.15 g/kg) |
|---|---|---|---|---|---|---|
| Kidney | 0.63±0.03 | 0.71±0.04## | 0.71±0.06 | 0.75±0.03 | 0.72±0.05 | 0.69±0.06 |
| Heart | 0.32±0.04 | 0.33±0.03 | 0.35±0.06 | 0.34±0.03 | 0.33±0.03 | 0.34±0.05 |
| Liver | 3.50±0.25 | 3.07±0.15## | 2.83±0.45 | 3.01±0.18 | 3.42±0.42* | 3.17±0.43 |
| Spleen | 0.18±0.04 | 0.20±0.02 | 0.35±0.15** | 0.20±0.03 | 0.20±0.03 | 0.19±0.02 |
| Lung | 0.53±0.07 | 0.67±0.12## | 0.76±0.07 | 0.63±0.10 | 0.60±0.08 | 0.57±0.10 |

*$P < 0.05$ and

**$P < 0.01$ compared with model group

#$P < 0.05$ and

##$P < 0.01$ compared with normal group.

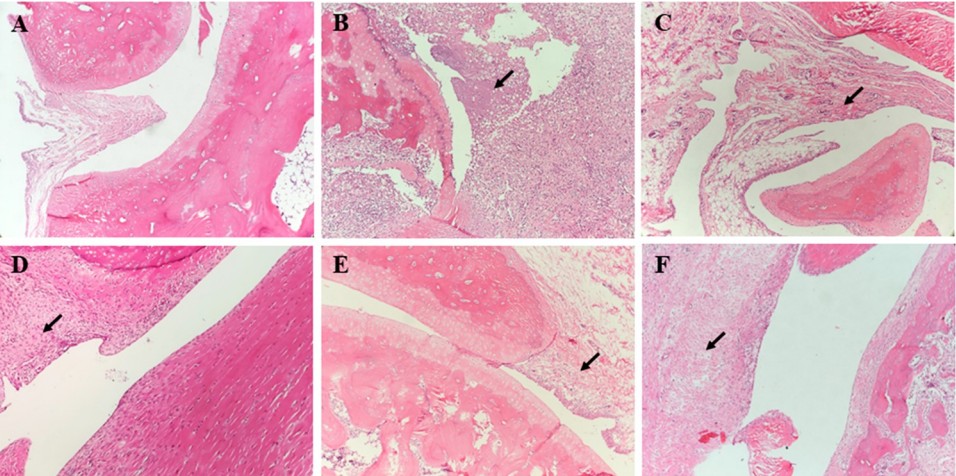

**Fig 5. Histopathological examination of ankle joints (×100).** (A) normal group, (B) model group, (C) methotrexate group, (D) *P. sinensis* extract group (0.6 g/kg), (E) *P. sinensis* extract group (0.3 g/kg), (F) *P. sinensis* extract group (0.15 g/kg).

isolation of *P. sinensis*; only nine compounds were reported from this plant [20]. Therefore, systematic studies on the phytochemistry of *P. sinensis* need to be carried out. Future phyto-chemical studies of *P. sinensis* should focus on separating and identifying distinct chemical components to clarify the chemical similarities and differences between *P. sinensis* and Ery-cibes Caulis.

The potential components and targets of *P. sinensis* against RA were analyzed using a net-work pharmacology approach, and five compounds, twenty-five targets, and eight pathways were finally obtained. Esculetin was shown to possess an anti-inflammatory effect by suppress-ing the HIF-1α signaling pathways [25]. Caffeic acid attenuated hepatocellular carcinoma cells' angiogenesis by reducing JNK-1-mediated HIF-1α stabilization [26]. Umbelliferone was useful for treating arthritis by suppressing the MAPK/NF-κB pathway [27]. In this study, KEGG analysis and C-T-P network showed that *P. sinensis* regulated the PI3K/Akt, HIF-1, Estrogen, Rap1 signaling pathways and so on. Recent studies demonstrated that PI3K/AKT pathway inhibits apoptosis in chondrocytes, and modulation of the pathway has been proposed as a

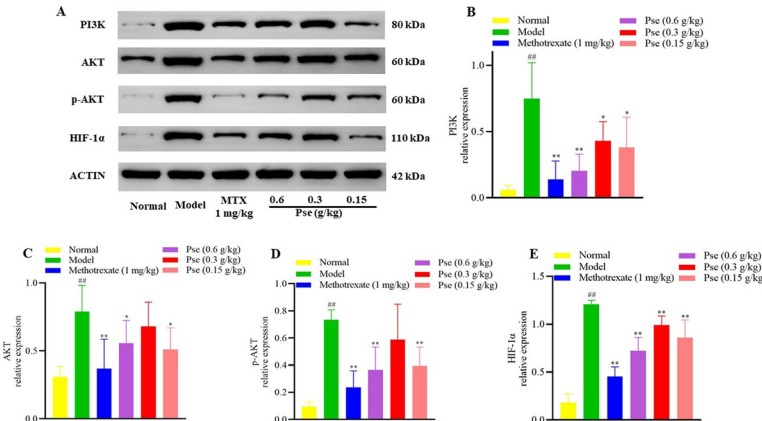

**Fig 6. Effects of *P. sinensis* extract (Pse) on the expression levels of PI3K, AKT, p-AKT, and HIF-1α proteins.** The specific bands of proteins (A), the expression levels of proteins (B~E).

potential therapy against RA [28,29]. HIF-1α can increase the production of inflammatory cytokines, and promote angiogenesis in RA patients [30,31]. The maintenance of Rap1 signaling in T cells can reduce the incidence rate and severity of CIA [32].

Macrophages play an important role in the pathogenesis of RA, and are the main source of inflammatory cytokines (such as NO, TNF-α, IL-1β, and IL-6) [33]. Therefore, we chose LPS-induced RAW 264.7 cells to verify the results of network pharmacology in vitro. A large number of literatures have reported the levels of TNF-α, IL-1β and IL-6 were significantly increased in serum of patients with RA [34]. At the same time, TNF is an important target of *P. sinensis* for the treatment of RA in our network pharmacology research. Therefore, these indexes were selected to verify the anti-inflammatory activity of Pse and five active compounds. It was reported that caffeic acid and caffeoylquinic acids could be active compounds of the anti-inflammatory potential of *P. sinensis* [7]. In this study, coumarins (esculetin, umbelliferone and scopolin) and amides (*trans-N*-feruloyltyramine) from *P. sinensis* showed good anti-inflammatory activities, especially amides, which should be attached great attentions to quality control of *P. sinensis* in the future.

Adjuvant-induced arthritis (AIA) and CIA models are classic animal models for RA research. However, AIA is different from RA in that it lacks a chronic pathological process [35]. At present, the CIA is recognized as the best RA model. We verified the computational prediction mechanisms of *P. sinensis* against RA based on the CIA rat model. The PI3K/AKT pathway is closely related to RA by deregulating activated immune cells' proliferation and synovial fibroblasts [28]. Zou et al [36] reported that HIF-1 levels in the peripheral serum of CIA rats positively correlated with AI, and the CIA rat model regulated the expression of HIF-1α proteins via PI3K pathway. Combined with network pharmacology and literature reports, the PI3K/AKT and HIF-1 pathway was selected for confirmation. The results suggested that Pse attenuates the severity, pathological changes, and the release of cytokines (IL-6 and HIF-1α) during RA progression in a dose-dependent manner. Western blot analysis demonstrated that Pse significantly reduced protein levels of PI3K, p-AKT, and HIF-1α in the inflamed joints of CIA rats. The experimental results are consistent with those of network pharmacology. As a positive control, methotrexate significantly inhibited the arthritic response of rats; however, the relative organ weight of the spleen was significantly higher than in rats of other groups, and the weight gain of rats was less than that of the model group, which may be side effects of methotrexate. By contrast, *P. sinensis* treated RA without these side effects. This suggests *P. sinensis* may be a better choice than methotrexate when treating RA.

In recent years, with the decrease of natural resources of Erycibes Caulis, *P. sinensis* has been used as the primary substitute. The study on the anti-rheumatic effect of *P. sinensis* provided evidence for its use as a substitute. The genus *Porana* is widely distributed globally; however, there are few medicinal uses. We expect this study on *P. sinensis* will accelerate the rational development and utilization of genus *Porana* plants.

## 5. Conclusion

We identified 21 compounds containing in *P. sinensis* by UPLC-MS and offered evidence that *P. sinensis* reverses the pathological events during RA progression by regulating the PI3K/AKT and HIF-1 signaling pathways. *P. sinensis* extract and five compounds (esculetin, umbelliferone, *trans-N*-feruloyltyramine, caffeic acid and scopolin) could inhibit the release of inflammatory mediators (NO, TNF-α, IL-1β and IL-6) in LPS-induced RAW 264.7 cell. These findings provide the experimental basis for the application of *P. sinensis* against RA. In addition, we expect that our findings may help develop *P. sinensis* into a substitute for Erycibes Caulis, thereby setting an example for the study of substitutes for TCM.

## Supporting information

**S1 Fig. The C-T-D network for *P. sinensis* in treatment of rheumatoid arthritis.**
(PDF)

**S2 Fig. The protein-protein interaction (PPI) network.**
(PDF)

**S3 Fig. The KEGG pathway for *P. sinensis* against rheumatoid arthritis.**
(PDF)

**S4 Fig. Effects of cytotoxicity on RAW264.7 cells of *P. sinensis* extract (Pse) and its effective constituents ($^{**}P < 0.01$).**
(PDF)

**S5 Fig.** Effects of *P. sinensis* extract (Pse) on ankle circumference (A), arthritis index (B) and body weight (C) of rats. Values shown are mean ± SD (n = 8); $^{*}P < 0.05$ and $^{**}P < 0.01$ compared with model group; $^{#}P < 0.05$ and $^{##}P < 0.01$ compared with normal group.
(PDF)

**S6 Fig. Effects of *P. sinensis* extract (Pse) on HIF-1α and IL-6 levels in collagen-induced arthritis rats.**
(PDF)

**S7 Fig.** Effects of *P. sinensis* extract (Pse) on mRNA levels of HIF-1α (A), PI3K (B) and AKT (C).
(PDF)

**S1 Table. Summary of gene-specific real-time PCR primer sequences.**
(DOC)

**S2 Table. The information of top 20 significant KEGG enrichment analysis.**
(DOC)

**S1 File.**
(ZIP)

## Acknowledgments

We are thankful to Dr. Xia Du from Shaanxi Academy of Traditional Chinese Medicine for providing technical assistance with network pharmacology.

## Author Contributions

**Data curation:** Jing Hu, Yuanyuan Yang, Tong Qu.

**Investigation:** Jing Hu, Lintao Zhao, Ning Li.

**Project administration:** Zhiyong Chen.

**Validation:** Hui Ren, Xiaomin Cui.

**Writing – original draft:** Zhiyong Chen.

**Writing – review & editing:** Hongxun Tao, Yu Peng.

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
