## [Decision Letter · Decision Letter 0]

21 Sep 2021

PONE-D-21-26996Investigation of the active ingredients and pharmacological mechanisms of Porana sinensis Hemsl. against rheumatoid arthritis using network pharmacology and experimental validationPLOS ONE

Dear Dr. Chen,

Thank you for submitting your manuscript to PLOS ONE. After careful consideration, we feel that it has merit but does not fully meet PLOS ONE’s publication criteria as it currently stands. Therefore, we invite you to submit a revised version of the manuscript that addresses the points raised during the review process.

Your manuscript has been reviewed by two experts in the field. Please provide a letter answering point-by-point the concenrs of the reviewers.In addition, please address the following: 1. The figures provided are very blurry.

 2. The level of the extract showed in Fig S5 are extremely high, and a toxicity assay in untreated animals should be shown.

3. Please provide the information related to the cytotoxicity, including the graphs.

We look forward to receiving your revised manuscript.

Kind regards,

Horacio Bach

Academic Editor

PLOS ONE

Journal Requirements:

"This work was supported by National Natural Science Foundation of China [grant numbers 81973419, 81603264]; Key Research and Development Program of Shaanxi [grant number 2020SF-328]; Shaanxi Administration of Traditional Chinese Medicine Projects [grant number 2021-PY-003]."

Reviewers' comments:

Reviewer's Responses to Questions

**Comments to the Author**

1. Is the manuscript technically sound, and do the data support the conclusions?

Reviewer #1: Yes

Reviewer #2: Yes

2. Has the statistical analysis been performed appropriately and rigorously? 

Reviewer #1: Yes

Reviewer #2: I Don't Know

3. Have the authors made all data underlying the findings in their manuscript fully available?

Reviewer #1: Yes

Reviewer #2: Yes

4. Is the manuscript presented in an intelligible fashion and written in standard English?

Reviewer #1: Yes

Reviewer #2: Yes

5. Review Comments to the Author

Reviewer #1: The manuscript entitle " Investigation of the active ingredients and pharmacological mechanisms of Porana sinensis Hemsl. against rheumatoid arthritis using network pharmacology and experimental validation " reviewed. The study design is well, but some points should be revised by authors:

- Authors should add more points about P. sinensis properties in introduction.

- Authors should add some points about RA in introduction.

- Authors should mention to “ Investigation of the mechanism of action of Porana sinensis Hemsl. against gout arthritis using network pharmacology and experimental validation” in references.

- Authors should add references for Methods in Method & Material part.

- Authors should explain more about groups of study in M&M.

- CIA as a model should be explained with refernces in M&M.

- Authors should add references for model, AI and duration of treatment in M&M.

- Authors should explain how selected the doses of treatment in the M&M or add references.

Reviewer #2: The manuscript entitled “Investigation of the active ingredients and pharmacological mechanisms of Porana sinensis Hemsl. against rheumatoid arthritis using network pharmacology and experimental validation” (PONE-D-21-26996) basically deals on if P. sinesis may act as a substitute of Erycibes Caulis.

I have only one suggestion for the authors: you could add the accuracy or probability that the compounds, that were not unambiguously identified, are the ones mentioned.

6. PLOS authors have the option to publish the peer review history of their article (what does this mean?). If published, this will include your full peer review and any attached files.

Reviewer #1: No

Reviewer #2: No

---

## [Author Response · Author response to Decision Letter 0]

21 Oct 2021

Manuscript Reference Number: PONE-D-21-26996R1

Manuscript Title: Investigation of the active ingredients and pharmacological mechanisms of Porana sinensis Hemsl. against rheumatoid arthritis using network pharmacology and experimental validation

Journal: PLOS ONE

Response to the editors’ comments:

1. The figures provided are very blurry.

Response: Thanks for your suggestion. The figures have been revised and uploaded to PACE digital diagnostic tool to ensure that figures meet PLOS ONE's requirements.

2. The level of the extract showed in Fig S5 are extremely high, and a toxicity assay in untreated animals should be shown.

Response: Thanks for your suggestion. The toxicity of 40% ethanolic extract of Porana sinensis has been evaluated in our previous work [1]. A single dose of the extract (5.0 g/kg) was administered (ig) to 10 mice (five males and five females). Behaviors such as hyperactivity, sedation, increased or decreased respiration, loss of righting reflex and food and water intake were observed over a period of 14 days. On day 14, all animals were sacrificed and subjected to necropsies. The results showed that all animals gained weight, appeared normal and the necropsy revealed no visible lesions in any animals. Thus, the oral LD50 values, for 40% ethanolic extract of P. sinensis, for female and male mice must be greater than 5.0 g/kg.

3. Please provide the information related to the cytotoxicity, including the graphs.

Response: Thanks for your suggestion. Cytotoxicity data have been provided in S4 Fig in Supporting Information.

Response: We ensure that our manuscript meets PLOS ONE's style requirements.

5. We note that the grant information you provided in the ‘Funding Information’ and ‘Financial Disclosure’ sections do not match. When you resubmit, please ensure that you provide the correct grant numbers for the awards you received for your study in the ‘Funding Information’ section. We note that you have provided funding information that is not currently declared in your Funding Statement. However, funding information should not appear in the Acknowledgments section or other areas of your manuscript. We will only publish funding information present in the Funding Statement section of the online submission form. Please remove any funding-related text from the manuscript and let us know how you would like to update your Funding Statement. Please include your amended statements within your cover letter; we will change the online submission form on your behalf.

Response: We have removed funding-related text from the manuscript, and provided funding information in the Funding Statement section.

6. PLOS ONE now requires that authors provide the original uncropped and unadjusted images underlying all blot or gel results reported in a submission’s figures or Supporting Information files. This policy and the journal’s other requirements for blot/gel reporting and figure preparation are described in detail at https://journals.plos.org/plosone/s/figures#loc-blot-and-gel-reporting- requirements and https://journals.plos.org/plosone/s/figures#loc-preparing -figures-from-image-files. When you submit your revised manuscript, please ensure that your figures adhere fully to these guidelines and provide the original underlying images for all blot or gel data reported in your submission. See the following link for instructions on providing the original image data: https://journals.plos.org/plosone/s/figures#loc-original-images-for-blots-and-gels. In your cover letter, please note whether your blot/gel image data are in Supporting Information or posted at a public data repository, provide the repository URL if relevant, and provide specific details as to which raw blot/gel images, if any, are not available.

Response: We have prepared our figures adhere to these guidelines and provided the original underlying images for all blot data in Supporting Information files (S1_raw_images.pdf).

Response to the reviewers’ comments:

Reviewer #1: The manuscript entitle "Investigation of the active ingredients and pharmacological mechanisms of Porana sinensis Hemsl. against rheumatoid arthritis using network pharmacology and experimental validation" reviewed. The study design is well, but some points should be revised by authors.

1. Authors should add more points about P. sinensis properties in introduction.

Response: Thanks for your suggestion. “P. sinensis, which belongs to the family Convolvulaceae, is mainly found in limestone mountainous regions and is widely distributed in China North-Central, China South-Central, China Southeast and Vietnam.”

2. Authors should add some points about RA in introduction.

Response: Thanks for your suggestion. “RA is a chronic autoimmune disease, which mainly acts on synovium, cartilage and bone, resulting in the decline of physical function and quality of life [9]. At present, nonsteroidal anti-inflammatory drugs (NSAIDs) and disease-modifying anti-rheumatic drugs (DMARDs) are commonly used in the treatment of RA. Although these drugs are typically effective, they are also not satisfactory because of their low efficacy and side effects [9]. It is of great significance to develop anti-RA TCM with multi-target effect and clear pharmacological effect.”

3. Authors should mention to “Investigation of the mechanism of action of Porana sinensis Hemsl. against gout arthritis using network pharmacology and experimental validation” in references.

Response: Thanks for your suggestion. This literature has been cited.

4. Authors should add references for Methods in Method & Material part.

Response: Thanks for your suggestion. References have been added.

5. Authors should explain more about groups of study in M&M.

Response: Thanks for your suggestion. This part has been rewritten. “The 48 rats were randomly divided into six groups: the normal, model, methotrexate (1 mg/kg), high-dose (Pse, 0.6 g/kg), middle-dose (Pse, 0.3 g/kg) and low-dose (Pse, 0.15 g/kg) groups. The selection of administered doses is based on our previous studies. At these doses, the extract shows good anti-inflammatory and analgesic effects [3].”

6. CIA as a model should be explained with references in M&M.

Response: Thanks for your suggestion. “The CIA model is one of the standard RA models, which shares several pathological features with RA, such as synovial inflammatory cell infiltration, synovial hyperplasia and bone erosion [9]. The CIA model was established as previously described [18].”

7. Authors should add references for model, AI and duration of treatment in M&M.

Response: Thanks for your suggestion. References have been added.

8. Authors should explain how selected the doses of treatment in the M&M or add references.

Response: Thanks for your suggestion. “The selection of administered doses is based on our previous studies. At these doses, the extract shows good anti-inflammatory and analgesic effects [3].”

Reviewer #2: The manuscript entitled “Investigation of the active ingredients and pharmacological mechanisms of Porana sinensis Hemsl. against rheumatoid arthritis using network pharmacology and experimental validation” (PONE-D-21-26996) basically deals on if P. sinensis may act as a substitute of Erycibes Caulis.

1. I have only one suggestion for the authors: you could add the accuracy or probability that the compounds, that were not unambiguously identified, are the ones mentioned.

Response: Thanks for your suggestion. Based on the high-accuracy precursor ions and product ions obtained from Q Exactive Focus MS, the elemental compositions were calculated. By searching literature sources, all components reported in the literature on P. sinensis and plants of the same family were summarized in a Microsoft Office Excel table to search the most rational molecular formula. The fragmentation patterns of these compounds were used to differentiate compounds with the same formula. References for compounds identification have been added in Table 1.

References:

1. Chen Z, Liao L, Zhang Z, Wu L, Wang Z. Comparison of active constituents, acute toxicity, anti-nociceptive and anti-inflammatory activities of Porana sinensis Hemsl., Erycibe obtusifolia Benth. and Erycibe schmidtii Craib. Journal of ethnopharmacology, 150(2): 501-506.

---

## [Decision Letter · Decision Letter 1]

13 Dec 2021

PONE-D-21-26996R1Investigation of the active ingredients and pharmacological mechanisms of Porana sinensis Hemsl. against rheumatoid arthritis using network pharmacology and experimental validationPLOS ONE

Dear Dr. Chen,

Thank you for submitting your manuscript to PLOS ONE. After careful consideration, we feel that it has merit but does not fully meet PLOS ONE’s publication criteria as it currently stands. Therefore, we invite you to submit a revised version of the manuscript that addresses the points raised during the review process. According to the new policies of PLoS One, authors should provide uncropped gels and blots. We have asked to provide this information, but now the raw data do not show the whole gels and blots. Please provide them.

Also, the cytotoxicity asked has not been fulfilled. Instead, the cytotoxicity of individual compounds has been included and not the extracts as mentioned.

We look forward to receiving your revised manuscript.

Kind regards,

Horacio Bach

Academic Editor

PLOS ONE

Reviewers' comments:

Reviewer's Responses to Questions

**Comments to the Author**

1. If the authors have adequately addressed your comments raised in a previous round of review and you feel that this manuscript is now acceptable for publication, you may indicate that here to bypass the “Comments to the Author” section, enter your conflict of interest statement in the “Confidential to Editor” section, and submit your "Accept" recommendation.

Reviewer #2: All comments have been addressed

2. Is the manuscript technically sound, and do the data support the conclusions?

Reviewer #2: Yes

3. Has the statistical analysis been performed appropriately and rigorously? 

Reviewer #2: N/A

4. Have the authors made all data underlying the findings in their manuscript fully available?

Reviewer #2: Yes

5. Is the manuscript presented in an intelligible fashion and written in standard English?

Reviewer #2: Yes

6. Review Comments to the Author

Reviewer #2: (No Response)

7. PLOS authors have the option to publish the peer review history of their article (what does this mean?). If published, this will include your full peer review and any attached files.

Reviewer #2: No

---

## [Author Response · Author response to Decision Letter 1]

16 Dec 2021

1. According to the new policies of PLoS One, authors should provide uncropped gels and blots. We have asked to provide this information, but now the raw data do not show the whole gels and blots. Please provide them.

Response: Thanks for your suggestion. We have provided uncropped gels and blots in S1_raw_images (pdf) in supporting information.

2. The cytotoxicity asked has not been fulfilled. Instead, the cytotoxicity of individual compounds has been included and not the extracts as mentioned.

Response: Thanks for your suggestion. The cytotoxicity of P. sinensis extract (Pse) has been provided in S4 Fig in supporting information.

---

## [Editor Report · Decision Letter 2]

13 Jan 2022

PONE-D-21-26996R2Investigation of the active ingredients and pharmacological mechanisms of Porana sinensis Hemsl. against rheumatoid arthritis using network pharmacology and experimental validationPLOS ONE

Dear Dr. Chen,

Thank you for submitting your manuscript to PLOS ONE. After careful consideration, we feel that it has merit but does not fully meet PLOS ONE’s publication criteria as it currently stands. Therefore, we invite you to submit a revised version of the manuscript that addresses the points raised during the review process.

We have sent two request to be addressed. The cropped gels are still an issue. You have provided the immunoblots already cropped from the original gel. The request is to show the original gel that was used to crop.

We look forward to receiving your revised manuscript.

Kind regards,

Horacio Bach

Academic Editor

PLOS ONE
---

## [Author Response · Author response to Decision Letter 2]

16 Feb 2022

1. We have sent two request to be addressed. The cropped gels are still an issue. You have provided the immunoblots already cropped from the original gel. The request is to show the original gel that was used to crop.

Response: Thanks for your suggestion. We have provided the original gel in S1_raw_images (pdf) in supporting information.

---

## [Editor Report · Decision Letter 3]

17 Feb 2022

Investigation of the active ingredients and pharmacological mechanisms of Porana sinensis Hemsl. against rheumatoid arthritis using network pharmacology and experimental validation

PONE-D-21-26996R3

Dear Dr. Chen,

We’re pleased to inform you that your manuscript has been judged scientifically suitable for publication and will be formally accepted for publication once it meets all outstanding technical requirements.

Kind regards,

Horacio Bach

Academic Editor

PLOS ONE
---

## [Editor Report · Acceptance letter]

22 Feb 2022

PONE-D-21-26996R3 

Investigation of the active ingredients and pharmacological mechanisms of *Porana sinensis* Hemsl. against rheumatoid arthritis using network pharmacology and experimental validation 

Dear Dr. Chen:

I'm pleased to inform you that your manuscript has been deemed suitable for publication in PLOS ONE. Congratulations! Your manuscript is now with our production department. 

Kind regards, 

on behalf of

Prof. Horacio Bach 

Academic Editor

PLOS ONE